# Integrated Single-Cell Transcriptome Analysis Reveals Novel Insights into the Role of Opioid Signaling in the Pathophysiology of Inflammatory Bowel Disease

**DOI:** 10.3390/biomedicines13061398

**Published:** 2025-06-06

**Authors:** Mudan Zhang, Zhuo Xie, Shenghong Zhang, Gaoshi Zhou

**Affiliations:** 1Department of Gastroenterology, The First Affiliated Hospital, Sun Yat-sen University, Guangzhou 510080, China; zhangmd23@mail2.sysu.edu.cn (M.Z.); xiezh27@mail2.sysu.edu.cn (Z.X.); 2Guangxi Hospital Division, The First Affiliated Hospital, Sun Yat-sen University, Nanning 530028, China

**Keywords:** single-cell transcriptomics, inflammatory bowel disease, opioid signaling, monocytes, therapeutic response

## Abstract

**Background/Objectives:** Inflammatory bowel disease (IBD), encompassing Crohn’s disease (CD) and ulcerative colitis (UC), is characterized by chronic gastrointestinal inflammatory diseases with complex etiology and remains a therapeutic challenge due to heterogeneous treatment responses. Opioids are widely used for analgesic management in IBD, yet the role of opioid signaling in IBD remains unclear. **Methods**: We employed single-cell RNA sequencing (scRNA-seq), bulk RNA sequencing, CellChat analysis, and transcription factor activity assessment to systematically investigate the roles and underlying mechanisms of opioid signaling-related genes in IBD. **Results:** We characterized opioid signaling-related genes in IBD at single-cell resolution and identified a novel subset of monocytes with a high expression level of opioid signaling-related genes (OpiHi monocytes). OpiHi monocytes were enriched in IBD tissues and served as a predominant source of tumor necrosis factor (TNF)-related signaling in the tissues of IBD. An inflammatory microenvironment in IBD drove the generation of OpiHi monocytes. Moreover, the prediction model based on OpiHi monocytes marker genes had robust predictive performance for the therapeutic response to anti-TNF therapy in IBD. **Conclusions**: This study provides novel insights into opioid signaling in IBD pathogenesis at the cellular level and establishes a reliable biomarker for precise management of anti-TNF therapy.

## 1. Introduction

Inflammatory bowel disease (IBD) are recurrent chronic inflammatory disorders of the gastrointestinal system, including two subtypes: Crohn’s disease (CD) and ulcerative colitis (UC) [1,2]. The etiology of IBD remains incompletely understood due to its multifaceted interplay among genetic susceptibility, environmental factors, microbiome dysfunction, and host immune response [3]. Over the past few decades, significant advancements in research have yielded multiple therapeutic modalities, such as small-molecular drugs, biologics, and immunomodulators [4]. Despite advancements in biologics, a wide subset of patients still failed to achieve sustained remission [5]. Therefore, identifying novel pathogenic components and prognostic markers is critical for optimizing the clinical management of IBD patients.

Abdominal pain is commonly reported in patients with inflammatory bowel disease (IBD), even in individuals who are in remission, which drives the widespread use of opioids for analgesia, with nearly 27% of patients becoming long-term users [6,7,8]. Previous studies have found that opioids have complex associations with the pathophysiological processes of IBD, which may not only alleviate inflammation through immune modulation, but also correlates with worsened clinical outcomes [9,10,11]. Recent studies have demonstrated that opioid use might be related to negative clinical outcomes in IBD, including higher emergency encounters and increased all-cause mortality [11,12,13,14]. In preclinical IBD mice experiments, administration of mu-opioid receptor (MOR) agonists or δ opioid agonists alleviated TNBS-induced murine colitis, whereas MOR-deficient mice showed higher vulnerability to colitis compared to the wild-type mice [10,15]. However, Wang et al. found that fentanyl exacerbated murine colitis by increasing the proportions of MOR ^+^ Th1 cells and macrophages [9]. Moreover, administration of naltrexone aggravated colitis-associated colorectal cancer [16]. Despite extensive research, whether opioids exert deleterious effects and the potential mechanisms by which opioid signaling affects IBD progression are still poorly understood.

In this study, we characterized opioid signaling-related genes in IBD at single-cell resolution and identified a novel subset of monocytes with a high expression level of opioid signaling-related genes (OpiHi monocytes). Moreover, we further evaluate the potential value of OpiHi monocytes in predicting the therapeutic response to anti-tumor necrosis factor (TNF) therapy in patients with IBD.

## 2. Materials and Methods

### 2.1. Single-Cell Transcriptome Sequence Data

We used a total of 75 ileum or colorectal biopsy samples from individuals with CD, UC, and normal controls (NC) across 4 GEO datasets (GSE134809 [17], GSE202052 [18], GSE214695 [19], and GSE231993 [20]). The expression matrices of these 4 single-cell transcriptome sequencing (scRNA-Seq) datasets were downloaded for further analysis. These datasets had already undergone upstream Cell Ranger count and aggregation analyses by their respective uploaders. Subsequent analyses were carried out in an R (version 4.3.3) environment, where single-cell Seurat objects were processed utilizing the Seurat package (version 5.1.0), and plots were generated utilizing the Tidyverse package (version 2.0.0) [21,22].

### 2.2. Quality Control and Removing Doublets

After merging all matrices of these 75 samples, quality control and removing doublets were executed successively. Single cells were qualified by the percentage of mitochondrial gene expression. Immune cells with mitochondrial gene expression below 5% were preserved, while stromal and epithelial cells with below 25% were preserved. The single cells express two types of markers among T cells, B/Plasma cells, myeloids, endothelial cells, epithelial cells, and fibroblasts at the same time, defined as doublets, and were removed from further analyses.

### 2.3. Cell Clustering, Annotating, and Proportion Calculation

Qualified singlet single cells were then clustered unsupervised through workflows of the Seurat package and annotated into 7 different major groups by their overexpressed marker genes. The cells annotated as myeloid cells were further integrated by the harmony package (version 1.2.1) to reduce batch effect, followed by being clustered and annotated into 8 subclusters by their marker genes. The cell proportions of myeloid subclusters were calculated through dividing the cell number of each subcluster by the total cell number of myeloid cells in each separated sample.

### 2.4. Opioid Signaling Evaluating and Function Enrichment Analysis

A gene set of opioid signaling genes, which is obtained from the REACTOME database [23], was used for the opioid signaling scoring of cells. All different types of cells, including immunocytes, stromal cells, and epithelial cells were scored by the AddModuleScore function in the Seurat package separately. These scores were represented as the opioid signaling indexes of cells while comparing between different groups in the same cluster or subcluster. To identify the function of OpiHi monocytes, Gene Ontology (GO) and Kyoto Encyclopedia of Genes and Genomes (KEGG) enrichment analysis was conducted through the clusterProfiler package (version 4.10.0) [24]. Marker genes of OpiHi monocytes and OpiLo monocytes were filtered using an adjusted *p* value less than 0.05, respectively.

### 2.5. Cell Communication Speculating

To speculate the communication between different cells, the CellChat package (version 2.1.2) was utilized to calculate the number of ligand–receptor pairs among stem cell and other cell clusters [25]. The pairs able to communicate within different locations (ileum and colorectum), and within different diseases (CD, UC, NC) were calculated, respectively. To avoid missing any possible pathways, “truncated Mean” type was used while communicating the probability of pathways, and the trim value was set to 0.1. The counts of communication numbers were used to summarize the cumulated communication strength between cell groups.

### 2.6. Cell Pseudotime Analysis

To speculate the derivation of monocytes, trajectory analysis was performed on all monocytes and macrophages in myeloid using the Monocle3 package (version 1.3.1) [26]. The developed skeleton, which included monocyte, was used to calculate the pseudotime of cells by setting monocyte as the beginning of progression. The cells on the skeleton, excluding monocytes, were not included in the pseudotime calculation.

### 2.7. Cell Transcription Factor Analysis

To find out the changed transcription factors of monocytes, the DecoupleR package (version 2.9.7) was used to find the transcription factors of these three subgroups [27]. The differential transcription factors were arranged in descending order of logarithms based on a 2-fold change in root sum square (RSS) values between the monocytes.

### 2.8. Chip Transcriptome Sequence Analysis and Clinical Prediction Model Building

To validate and extend the conclusion from single-cell datasets, we used three in-chip transcriptome sequence (bulk RNA-Seq) datasets from GEO (GSE12251 [28], GSE16879 [29], GSE23597 [30]), which included 132 intestinal samples from 36 CD and 96 UC patients with anti-TNF treatment. In GSE12251 accession, response to infliximab was defined as endoscopic and histologic healing at week 8. In GSE116879 accession, the patients were classified for response to infliximab based on endoscopic and histologic findings at 4–6 weeks after first infliximab treatment. In GSE23597 accession, response to infliximab was defined as endoscopic and histologic healing at week 8. Marker genes of OpiHi monocytes in CD and UC patients were used to calculate a score for each sample by the ssGSEA method from the GSVA package (version 1.50.0) [31]. The final logistic clinical prediction models and its receiver operating characteristic (ROC) curves were built by the opioid signaling ssGSEA score and anti-TNF treatment response through the pROC package (version 1.18.5) [32].

### 2.9. Statistical Analyses

Statistical analyses were conducted utilizing R software (version 4.3.3). Independent sample *t*-tests and Mann–Whitney U-tests were used to assess statistical differences in the data, which are expressed as mean ± SEM. Differences were considered statistically significant for *p*  <  0.05 (*), *p*  <  0.01 (**), *p*  <  0.001 (***), and *p* < 0.0001 (****).

## 3. Results

### 3.1. The Cellular Landscape of IBD Identified by scRNA-seq

To explore opioid signaling differences between IBD and normal intestinal tissues at single-cell resolution, a total of 75 ileum or colorectal single-cell transcriptome sequence (scRNA-Seq) samples among CD, UC, and NC patients from 4 GEO datasets [17,18,19,20] were integrated and re-analyzed. After quality control and removing low-quality cells and doublets (Appendix A), a total of 165,618 qualified cells were clustered and annotated into seven major groups: T cell, B cell, plasma cell, myeloid, fibroblast, endothelium, and epithelium (Figure 1A–C). The proportions of each cell type in each group are displayed in Figure 1D, and the canonical marker genes for seven major cell categories are shown in Figure 1E. As demonstrated, there were significant differences in the percentages of predominant cell populations among the groups. All the marker genes defining these cell types are displayed in Umap (Appendix A).

### 3.2. The Expression of Opioid Signaling-Related Gene Is Significantly Increased in Monocytes from IBD Tissues

Based on the gene list collected via the REACTOME database, a gene set of opioid signaling was used to score each cell group and evaluate the activity of opioid signaling pathways, respectively (Appendix A). We identified a significant upregulation of the opioid signaling-related genes in myeloid cells within the intestine of IBD patients (Appendix A). Further investigation into the opioid signaling-related genes across distinct subpopulations of myeloid cells revealed that monocytes from IBD samples showed a significantly higher opioid signaling score compared to NC, regardless of whether the samples were from CD or UC patients and irrespective of whether they originated from the ileum or colorectum (Figure 2A,B). These results indicate that opioid signaling is enhanced in IBD patients, with a high expression of opioid signaling-related genes observed in monocyte cells. To further investigate the expression patterns of opioid signaling in monocytes, all 7348 myeloid cells were clustered into four types (Figure 2C–E), including monocyte, macrophage, neutrophil, and dendritic cell (DC). The canonical marker genes for eight major cell groups are displayed in Figure 2F. Monocytes were separated into two groups based on the expression levels of opioid signaling-related genes: OpiHi monocytes and OpiLo monocytes. The marker genes of OpiHi monocytes are provided in Appendix A. OpiHi monocytes were significantly more prevalent in IBD tissues, whereas the abundance of OpiLo monocytes exhibited no significant difference between IBD tissues and NC tissues (Figure 2G,H). It is highly recommended that the opioid signaling alterations in myeloid cells observed in IBD be attributed to the increased proportion of OpiHi monocytes. The expression patterns of opioid receptor subtypes including μ-opioid receptor (MOR), δ-opioid receptor (DOR], κ-opioid receptor (KOR), and opioid receptor-like 1 (ORL-1) were further investigated across diverse cell types. Our results demonstrated that MOR was specifically expressed in T cells, ORL-1 exhibited broad expression in myeloid cells, and DOR showed high expression levels in endothelial cells (Appendix A). Otherwise, DCs were also clustered into classical dendritic cell 1, classical DC 2, plasmacytoid DC, and activated DC with LAMP3 expression, according to their different functions and marker genes [33,34,35,36]. The marker genes of all of the eight subgroups are shown in Umap (Appendix A).

### 3.3. TNF-Related Signaling Pathway Is Significantly Upregulated in OpiHi Monocytes

To further explore the function of OpiHi monocytes, we analyzed the differentially expressed genes (DEGs) between the OpiHi monocytes and OpiLo monocytes, and the results were displayed utilizing a volcano plot (Figure 3A). GO and KEGG enrichment analysis were performed on the DEGs between the two groups (Figure 3B,C). OpiHi monocytes exhibited significant up-regulated inflammation pathway genes, particularly the tumor necrosis factor (TNF)-related signaling pathway. In contrast, OpiLo monocytes displayed typical innate immune functions, including bacterial processing and antigen presentation. A deeper analysis of the differences in cytokine genes between OpiHi monocytes and OpiLo monocytes demonstrated that OpiHi monocytes are distinguished by their secretion of inflammatory factors associated with MIF, TNF, TNFSF10, and TNFSF13B. Additionally, we conducted a systematic screening for potential flow cytometry protein markers specific to OpiHi monocytes, identifying LCP2, TNFRSF1B, and GNA15 as promising candidates for cell surface marker detection in flow cytometry (Appendix A). To elucidate the intercellular communication mechanisms between OpiHi monocytes and other cell types, CellChat analysis was conducted (Figure 3D–G). To further identify the changed pathways from OpiHi monocytes to other cells, we analyzed and summarized all the changed signal pathways from OpiHi monocytes to other cells in IBD to NC samples (Appendix A). A variety of inflammatory factors, especially TNF-related signals, were significantly enhanced in both CD and UC compared to NC (Figure 3H).

### 3.4. The Inflammatory Microenvironment in IBD Tissues Drives the Generation of OpiHi Monocytes

Pseudotime trajectory analysis was performed between all macrophages and monocytes to reveal the derivation of monocytes in IBD tissues. It was shown that monocytes exhibit differentiation to macrophages (Figure 4A). Also, pseudotime trajectory analysis revealed that the expression of inflammation cytokines such as TNF and IL-1B were decreased during the differentiation of monocytes towards macrophages (Figure 4B). Furthermore, CellChat analysis revealed that multiple cell types, such as T cells, actively transmit inflammatory signals to OpiHi monocytes. (Figure 4C). To investigate the regulatory factors that play crucial roles in the generation of OpiHi monocytes in IBD, changes in transcription factors between IBD monocytes and NC monocytes, as well as between OpiHi monocytes and OpiLo monocytes, were analyzed (Appendix A). Compared with monocytes in NC tissues, monocytes in IBD tissues demonstrate enhanced activity of inflammation-related transcription factors. Notably, OpiHi monocytes exhibit significantly higher activity of key inflammation-related transcription factors, including STAT1, STAT3, and HIF1A, compared to OpiLo monocytes (Figure 4D,E, Appendix A). These findings collectively indicate that the intestinal inflammatory microenvironment plays a critical role in driving the generation of OpiHi monocytes.

### 3.5. OpiHi Monocytes Signature Gene Can Predict the Response of Anti-TNF Therapy

Given that OpiHi monocytes serve as a major source of TNF in IBD tissues, we further investigated the association between OpiHi monocytes and the response to anti-TNF treatment in IBD. Therefore, gene expression profiling data derived from 132 intestinal biopsy specimens collected from IBD patients prior to anti-TNF therapy were utilized to evaluate the predictive efficacy of OpiHi monocytes for treatment response (Appendix A). Prediction models were built by scoring OpiHi monocyte marker genes of each sample for CD and UC, respectively. The area under the curve (AUC) of the models built by different numbers of top marker genes of OpiHi monocytes are shown in Appendix A. For CD, the optimal prediction model was built using the top 40 OpiHi monocytes marker genes, achieving an AUC of 0.966. For UC, the optimal model was constructed using the top 80 marker genes, achieving an AUC of 0.822 (Figure 5A,B). Additionally, the same model was further applied to another GEO dataset (GSE234736 [37]) to predict the response to vedolizumab treatment in IBD. The results demonstrated that the model lacked sufficient predictive power for evaluating the therapeutic response to vedolizumab therapy in IBD (Figure 5C,D, Appendix A).

## 4. Discussion

Recent research advances have highlighted the significance of opioid signaling in cancer progression [38] and inflammatory diseases [39,40]. To elucidate the integrated roles of opioid signaling in IBD, we systematically characterized the expression of opioid signaling-related genes in IBD at single-cell resolution and identified a distinct subset of inflammatory monocytes (OpiHi monocytes). OpiHi monocytes exhibited molecular features of an increased expression of opioid signaling-related genes and were significantly enriched in IBD patients. In OpiHi monocytes, several pathways associated with inflammation, especially TNF signaling pathways, were upregulated. The OpiHi monocytes model demonstrated robust prognostic value for anti-TNF therapy responses (AUC = 0.966 in CD; AUC = 0.822 in UC). Collectively, through integrated analysis of scRNA-seq and bulk RNA-seq data, we identified a subset of inflammatory monocytes, termed OpiHi monocytes, which were marked by high expression of opioid signaling-related genes. Based on signature genes from this subgroup, we established a predictive model for the anti-TNF response in IBD patients. Our findings offer fresh perspectives on the investigation of opioid signaling in IBD.

Opioid signaling pathways, mediated by four G protein-coupled receptors, specifically μ-opioid receptor (MOR), δ-opioid receptor (DOR], κ-opioid receptor (KOR), and opioid receptor-like 1 (ORL-1), are classically associated with pain modulation but are increasingly recognized for their immunomodulatory roles [41,42]. These receptors are mainly expressed on neurons, ectodermal cells, and immune cells, including monocytes and macrophages [39]. When an endogenous or exogenous opioid agonist attaches to the receptor, the Gαi/o protein binds to GTP instead of GDP and leads to the dissociation of Gα-GTP from the Gβγ subunits and subsequently alters the downstream intracellular signaling cascade [41]. Researchers have demonstrated that opioids have complex interplay with the pathophysiological processes of IBD [9,10,11]. MOR agonists have been demonstrated to alleviate colitis in preclinical models of IBD by dampening inflammatory cytokine production, including TNF-α [10], while chronic opioid use in IBD patients correlates with adverse clinical outcomes [11]. The seemingly paradoxical effects may stem from receptor-specific signaling dynamics, microenvironmental clues, or interactions with other inflammatory pathways such as TNF-α. Our identification of OpiHi monocytes underscores the need to dissect how opioid receptor activation in specific immune subsets affects IBD pathogenesis. The identification of OpiHi monocytes suggests a previously unrecognized mechanism whereby opioid signaling in specific immune subsets may amplify TNF-mediated inflammation, potentially explaining the clinical dichotomy between protective receptor agonism and detrimental long-term opioid use in IBD.

As pivotal mediators of the immune system, monocytes exhibit a dual role in the pathogenesis of inflammatory bowel disease (IBD) by dynamically modulating the balance between intestinal inflammation and tissue repair [41]. Investigating the regulatory dynamics and mechanisms underlying monocyte differentiation and function may reveal novel therapeutic targets for IBD. A recent study demonstrated that tissue monocytes were markedly increased in active IBD compared to quiescent IBD and healthy controls, with these monocytes identified as the primary cellular source of IL-1β production [42]. Furthermore, Mitsialis et al. [43] confirmed from a single-cell perspective that the expression of IL-1β^+^ macrophages/monocytes was upregulated in both the active IBD mucosa and peripheral blood. Notably, researchers have found that IL-1α/IL-1β and IL-10 are critical regulators of monocyte IL-23 production, distinguishing homeostatic IL-23 production from inflammation-associated IL-23 production in patients with severe ulcerative active Crohn’s disease [44]. However, the functional heterogeneity of monocyte subsets at the single-cell level remains incompletely characterized. In this study, we identified a previously uncharacterized subset of inflammatory monocytes, termed OpiHi monocytes. By conducting differential gene expression analysis, we demonstrated that these cells are distinguished by their secretion of MIF, TNF, TNFSF10, and TNFSF13B, as well as their involvement in regulating the TNF signaling pathway.

To decipher the intercellular crosstalk, we employed CellChat analysis, which revealed significantly enhanced communication from OpiHi monocytes to other cell types in IBD tissues at single-cell level. In IBD tissues, we observed that cell communications from OpiHi monocytes to other cells were significantly elevated in IBD compared to NC. More importantly, compared with NC, OpiHi monocytes in IBD tissues transmitted a variety of inflammatory signals to multiple cell types. Notably, the TNF-related signaling pathway was significantly activated in the intercellular communication mediated by OpiHi monocytes. Our findings strongly indicated that OpiHi monocytes are expanded in IBD, transmitting more inflammatory signals, particularly TNF-related signaling pathways to neighboring cells, thereby contributing to a pro-inflammatory immune microenvironment in IBD.

We further investigated the origin of OpiHi monocytes in IBD tissues. By employing pseudotime trajectory analysis, we elucidated the differentiation trajectories of monocytes into macrophages. Notably, as monocytes progressively differentiated into macrophages, the expression levels of inflammatory cytokines, such as TNF and IL-1β, were significantly diminished. Concurrently, CellChat analysis demonstrated that stromal cells in IBD tissues, including immune cells, fibroblasts, and endothelial cells, transmit a considerable amount of inflammatory signaling to OpiHi monocytes. These findings suggest that the emergence of OpiHi monocytes may result from the influence of an imbalanced immune microenvironment in IBD on monocytes. To substantiate this hypothesis, we performed an analysis of transcription factor activity differences between NC monocytes and IBD monocytes, as well as between OpiHi monocytes and OpiLo monocytes. Through transcription factor activity analysis, we observed that OpiHi monocytes exhibit markedly higher activity of inflammation-related transcription factors, including STAT1, STAT3, and HIF1A, compared to OpiLo monocytes. Collectively, these data strongly indicate that the inflammatory microenvironment in IBD tissues plays a pivotal role in the generation of OpiHi monocytes in IBD tissues.

Given that TNF has been proved as a pivotal inflammatory cytokine in the pathogenesis and progression of IBD [43], several types of anti-TNF antibodies have been established as treatment protocols for IBD [44]. However, only a subset of patients experienced sustained long-term benefits from anti-TNF therapy in clinical practice [43]. Certain biomarkers, such as gene polymorphisms and gut microbiota [45], have demonstrated significant value in predicting the response to anti-TNF therapy in IBD patients; however, their predictive efficacy remains suboptimal. Single-cell sequencing analysis provides a novel approach for predicting TNF responsiveness in IBD patients [17]. Our findings indicate that OpiHi monocytes are strongly associated with the activation of the TNF signaling pathway in IBD patients. Therefore, we hypothesize that OpiHi monocytes may be associated with the therapeutic response to anti-TNF therapy in IBD. Our results showed that patients with lower OpiHi monocyte scores were found to have a favorable response to anti-TNF treatment, suggesting that the OpiHi monocyte score may be a reliable model to assess IBD patient eligibility for anti-TNF therapy. Moreover, this model was unable to predict the therapeutic response to vedolizumab in IBD patients, further highlighting the specific role of OpiHi monocytes in the activation of the TNF signaling pathway in IBD patients. It is suggested that OpiHi monocytes represent an effective factor for predicting the clinical response to anti-TNF treatment in IBD patients, which could assist physicians in determining whether to use anti-TNF therapy for IBD patients in clinical practice. This is the first study to analyze the expression levels of opioid signaling-related genes in IBD at single-cell resolution, thereby enhancing our understanding of the pathophysiological mechanisms and potentially informing clinical decision-making in IBD.

While our study provides novel insights, several limitations should be noted. First, functional uniqueness validation of OpiHi monocytes using primary cell cocultures and flow cytometry was precluded by technical constraints. Second, the bulk RNA datasets used for validation lacked detailed clinical data, including heterogeneity in disease severity and concurrent medications, which potentially confound the association between OpiHi monocyte signature and treatment outcomes. Therefore, future studies integrating single-cell multiomics with prospective phenotyping are warranted.

## 5. Conclusions

In the present study, we analyzed the expression levels of opioid signaling-related genes in IBD at single-cell resolution and identified a distinct subset of inflammatory monocytes (OpiHi monocytes). OpiHi monocytes were expanded in IBD and transmitted inflammatory signals, particularly TNF-related signaling, to neighboring cells, thereby contributing to the pro-inflammatory immune microenvironment in IBD. Based on the expression profiles of OpiHi monocytes using ssGSEA, we developed an optimized OpiHi monocyte model for predicting the therapeutic response to anti-TNF therapy in IBD patients. Collectively, these findings have significant implications for elucidating the pathophysiological mechanisms and advancing precision medicine in IBD.

## Figures and Tables

**Figure 1 biomedicines-13-01398-f001:**
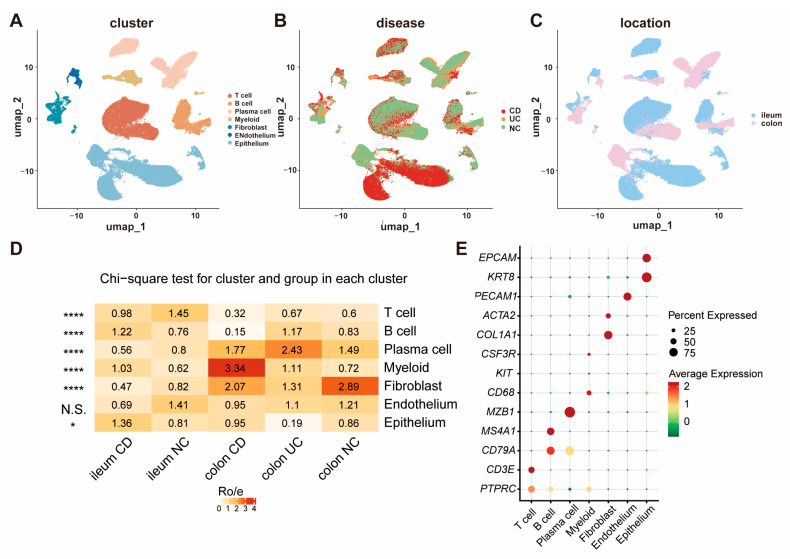
The cellular landscape of IBD identified by scRNA-seq. (**A**) UMAP visualization of 165,618 cells grouped into seven distinct clusters; (**B**) cellular distribution across CD, UC, and NC samples; (**C**) comparison of cell distribution between ileum and colorectum tissues; (**D**) proportions of each cell type within each group; (**E**) canonical marker genes for the seven major cell clusters. (* *p* < 0.05, **** *p* < 0.0001).

**Figure 2 biomedicines-13-01398-f002:**
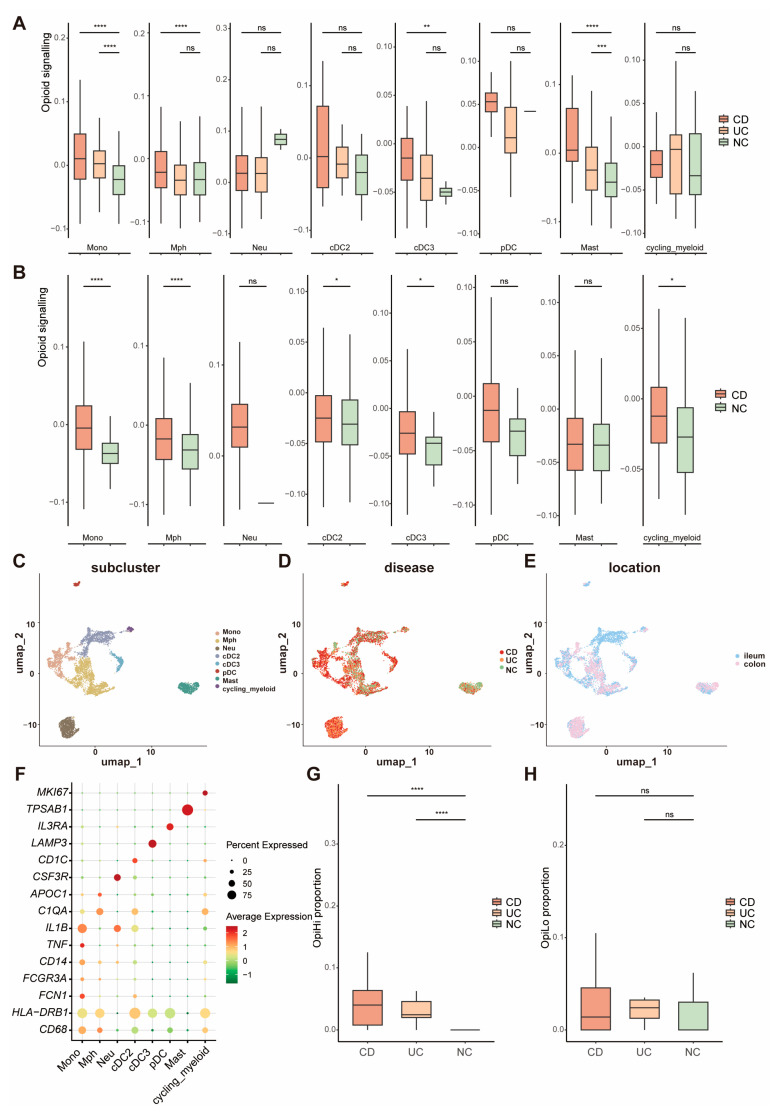
The expression of opioid signaling-related genes is significantly increased in monocytes from IBD tissues. (**A**,**B**) Opioid signaling scores of distinct subpopulations of myeloid cells across UC and CD in colorectum and ileum; (**C**) UMAP visualization of 7,348 cells grouped into eight distinct clusters; (**D**) cellular distribution among CD, UC, and NC samples; (**E**) distribution of the cells between ileum and colorectum; (**F**) marker genes of each subgroup of myeloid cells; (**G**,**H**) proportions of OpiHi monocytes and OpiLo monocytes across UC and CD; (**G**) OpiHi monocytes exhibited a significant increase in IBD tissues; (**H**) the abundance of OpiLo monocytes exhibits no significant differences between IBD tissues and NC tissues. (* *p* < 0.05, ** *p* < 0.01, *** *p* < 0.001, **** *p* < 0.0001).

**Figure 3 biomedicines-13-01398-f003:**
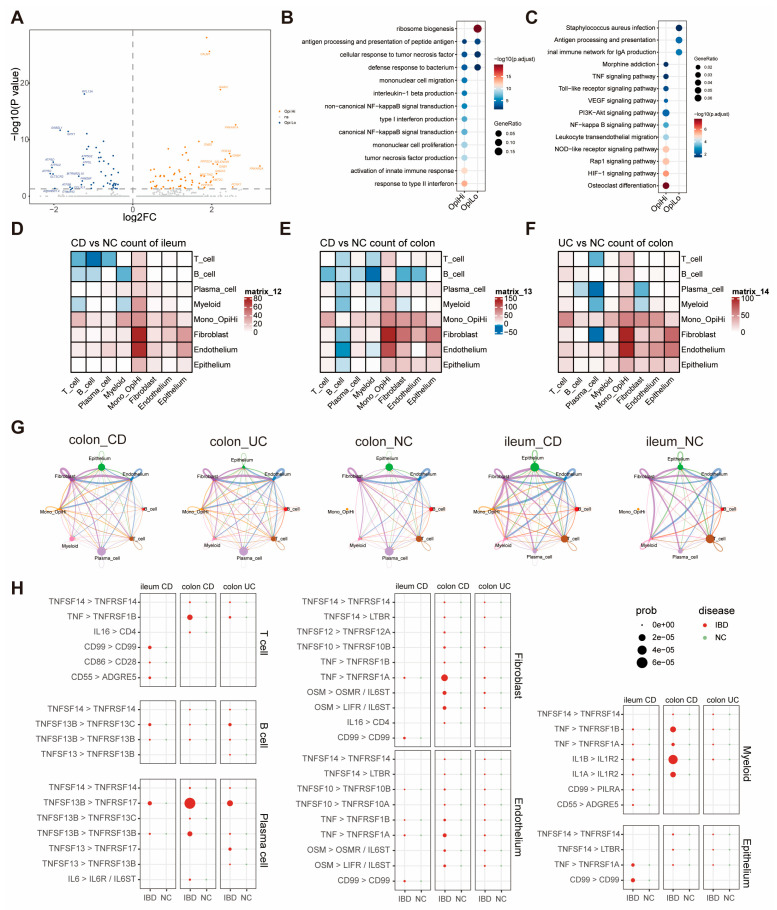
TNF-related signaling pathway is significantly upregulated in OpiHi monocytes. (**A**) Differentially expressed genes (DEGs) between the OpiHi monocytes and OpiLo monocytes were visualized by volcano plot, with log_2_ fold change (log_2_FC) on the x-axis and −log_10_(*p* value) on the y-axis. The horizontal dashed line indicates the significance threshold (*p* < 0.05); (**B**) the differences in GO functions between OpiHi monocytes and OpiLo monocytes; (**C**) the differences in KEGG enrichment functions between OpiHi monocytes and OpiLo monocytes; (**D**–**F**) the communication signal numbers differences between IBD and NC, respectively. Red shows the signal numbers increased in IBD, blue shows increased in NC, and white shows no changes between IBD and NC; (**G**) the communication signal intensity differences between IBD and NC; (**H**) the increased inflammation signal pathways from OpiHi monocytes to other cells in IBD versus in NC.

**Figure 4 biomedicines-13-01398-f004:**
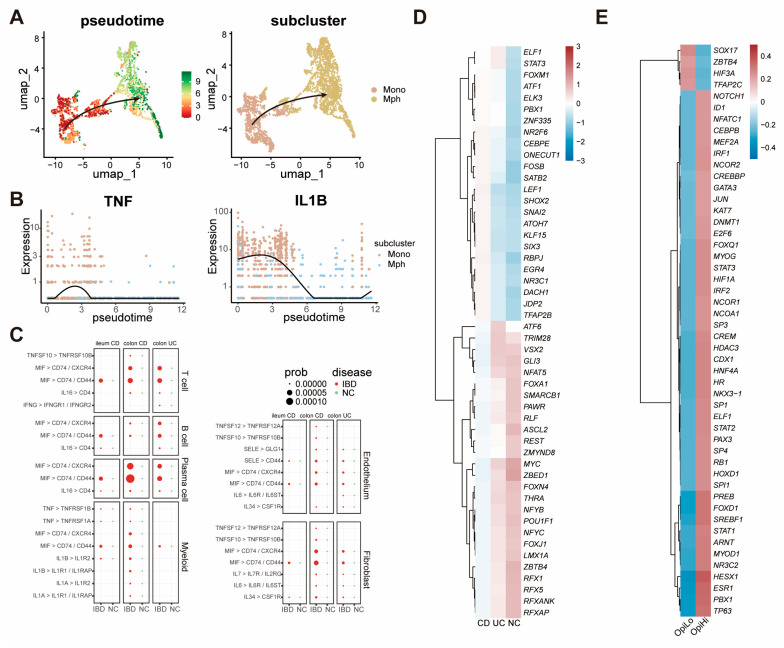
The inflammatory microenvironment in IBD tissues drives the generation of OpiHi monocytes. (**A**) Pseudotime trajectories illustrating the differentiation of monocytes into macrophages; (**B**) the expression of TNF and IL1B genes during the differentiation of monocytes towards macrophages; (**C**) the increased inflammation signal pathways from stromal cells to OpiHi monocytes in IBD versus in NC; (**D**) differential transcription factor activity between IBD monocytes and NC monocytes; (**E**) differential transcription factor activity between OpiHi monocytes and OpiLo monocytes.

**Figure 5 biomedicines-13-01398-f005:**
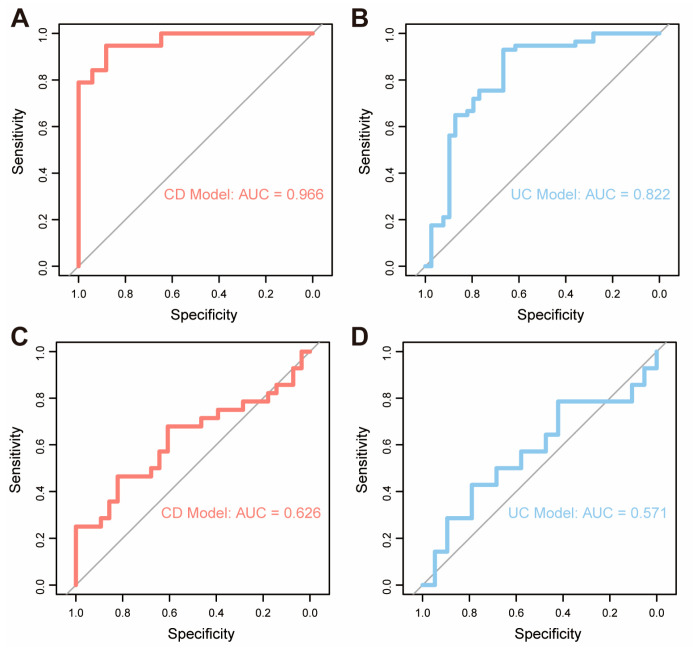
OpiHi monocytes signature gene can predict the response of anti-TNF therapy. (**A**) ROC curve analysis of the prediction model built by characteristic genes of OpiHi monocytes for evaluating the therapeutic response to anti-TNF therapy in CD; (**B**) ROC curve analysis of the prediction model constructed by characteristic genes of OpiHi monocytes for evaluating the therapeutic response to anti-TNF therapy in UC; (**C**) ROC curve analysis of the prediction model built by characteristic genes of OpiHi monocytes for evaluating the therapeutic response to vedolizumab therapy in CD; (**D**) ROC curve analysis of the prediction model constructed by characteristic genes of OpiHi monocytes for evaluating the therapeutic response to vedolizumab therapy in UC.

## Data Availability

The datasets of scRNA-Seq and bulk RNA-Seq are all available from GEO dataset “https://www.ncbi.nlm.nih.gov/geo (accessed on 28 March 2025)”. The codes for all procedures in R of this article can be obtained by contacting the corresponding author.

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
