# Peer review of "Integrated Single-Cell Transcriptome Analysis Reveals Novel Insights into the Role of Opioid Signaling in the Pathophysiology of Inflammatory Bowel Disease"

_biomedicines, 2025, doi:10.3390/biomedicines13061398_

Round 1

Reviewer 1 Report

Comments and Suggestions for Authors

we read with interest the article titled Integrated Single-cell Transcriptome Analysis Reveals Novel Insights into the Role of Opioid Signaling in the Pathophysiology of Inflammatory Bowel Disease. in this work the PIs have utilized multi-omics approaches involving single-cell RNA sequencing (scRNA-seq), bulk RNA sequencing, CellChat analysis, and transcription factor activity assessment to study the roles and underlying mechanisms of opioid signaling-  in IBD.

there are some concerns as described below:

Comments

the study carries novelty as the identification of a monocyte subset with high expression of opioid signaling-related genes (OpiHi monocytes) and its enrichment in IBD is potentially novel; however the PIs need to clarify the the uniqueness and functional distinction of OpiHi monocytes compared to other known inflammatory monocyte subsets in IBD.

2- Provide the list of genes in the "opioid signalling gene set"  to allow assessment of its specificity to opioid signaling versus general inflammation.

3-Demonstrate how OpiHi monocytes are distinct from previously described pro-inflammatory monocytes in IBD (e.g., S100A8/A9+, IL1B+ populations), ensuring it's not a re-labelling of a known signature.

4-  Include the detailed cohort information for the 132 samples (CD vs. UC, responders vs. non-responders per dataset) and   define "anti-TNF treatment response."

Results weakness: 

There should be additional experiments to provide stronger evidence for the functional uniqueness of the OpiHi monocytes beyond the gene signature. Consider if unique surface markers could enable flow cytometry validation. 

-Mechanistically, the authors should investigate whic and evaluate their secretome if possible (inflammatory cytokine production and signaling pathways).

Minor comments:

  • The overall scientific tone is generally appropriate. However, typos, punctuation, and grammar need to be proofread. Minor typos, punctuation errors, and consistent terminology (e.g., "opioid signalling" vs. "opioid signaling") are recommended throughout the manuscript; I also notice that the authors should use some transitions between paragraphs to ensure a smooth narrative flow.

Reviewer 2 Report

Comments and Suggestions for Authors

Date: April 1, 2025
Manuscript Title: Integrated Single-cell Transcriptome Analysis Reveals Novel Insights into the Role of Opioid Signaling in the Pathophysiology of Inflammatory Bowel Disease
Journal: Biomedicines

This manuscript presents an innovative and methodologically sound investigation into the role of opioid signaling in inflammatory bowel disease (IBD) using an integrative approach that combines single-cell transcriptomics, bulk RNA-sequencing, CellChat communication analysis, transcription factor profiling, and predictive modeling. The central contribution of the study is the identification and characterization of a novel monocyte subpopulation, termed OpiHi monocytes, which exhibit enriched expression of opioid signaling-related genes and significant involvement in TNF-related inflammatory pathways in IBD. The authors further demonstrate the translational relevance of this finding by constructing a predictive model for anti-TNF therapy response based on OpiHi monocyte signature genes.

The introduction effectively sets the stage by highlighting the clinical burden of IBD and the paradoxical role of opioid use in its management. The rationale for investigating opioid signaling at the single-cell level is compelling, and the authors frame their objectives clearly in light of existing clinical uncertainty surrounding opioids’ immunomodulatory roles.

The methods section is detailed and reproducible, employing appropriate computational tools for data integration (Seurat, Harmony), trajectory inference (Monocle3), intercellular communication (CellChat), transcription factor activity analysis (DecoupleR), and predictive modeling (ssGSEA, pROC). The use of multiple public GEO datasets increases the robustness and generalizability of the findings. One strength of the study is the stringent quality control, including careful doublet removal and thresholding based on mitochondrial gene content. This enhances confidence in the downstream cell-type annotations.

The results are comprehensive and well-illustrated. The discovery of OpiHi monocytes is clearly supported by scRNA-seq clustering and opioid signaling scoring, with further functional characterization using enrichment analyses and cell-cell communication profiles. The finding that these monocytes are major contributors to TNF signaling in IBD provides a mechanistic explanation for the observed pro-inflammatory effects of opioids in clinical contexts. Moreover, the use of transcription factor activity profiles, particularly the upregulation of STAT1, STAT3, and HIF1A, reinforces the inflammatory phenotype of OpiHi monocytes and ties in well with pseudotime analyses illustrating their derivation within the IBD microenvironment.

The translational arm of the study is especially noteworthy. The predictive model for anti-TNF response, constructed using OpiHi monocyte signature genes, demonstrates strong diagnostic performance in both Crohn’s disease and ulcerative colitis cohorts (AUC > 0.8). Importantly, the specificity of the model to anti-TNF therapy—not vedolizumab—underscores the biological relevance of the TNF axis in OpiHi monocytes, which strengthens the argument for their potential as a biomarker.

Nonetheless, a few areas could benefit from additional clarity or discussion. The manuscript would be strengthened by a more explicit discussion of how opioid receptor subtypes (MOR, DOR, KOR) are differentially expressed across immune cell types. Although the authors refer to preclinical models suggesting diverse effects of receptor agonism, the current manuscript does not clearly show whether specific receptors drive the transcriptional profile of OpiHi monocytes. Furthermore, while the predictive models are promising, the authors should comment on potential confounders such as disease severity or concurrent medications in the bulk RNA datasets, which may influence both monocyte profiles and treatment outcomes.

Minor typographical errors are present throughout the manuscript (e.g., “conut” instead of “count” in Figure S1 and inconsistent formatting in figure legends), and a light round of editorial proofreading is recommended to improve polish. The supplementary figures are useful and visually support the main text, but some of the legends would benefit from expanded explanations, especially in Figure S3 and S4, for readers unfamiliar with UMAP visualization conventions.

In conclusion, this is a timely and high-impact study that makes a meaningful contribution to both the mechanistic understanding and clinical management of IBD. The identification of OpiHi monocytes as key drivers of opioid-associated immune modulation in IBD provides a novel cell-type-specific biomarker and therapeutic target. The integration of single-cell data with clinical outcome modeling sets a valuable precedent for precision medicine in autoimmune and inflammatory diseases. I recommend this manuscript for publication pending minor revision to address the clarifications and proofreading suggestions mentioned above.

Round 2

Reviewer 1 Report

Comments and Suggestions for Authors

Comments are satisfactory